# (Un)bounding the Meta-Organization: Co-Evolution and Compositional Dynamics of a Health Partnership

**Steve Cropper [1,*] and Sanne Bor [2]**

[1] Research Office, Faculty of Humanities and Social Sciences, Keele University, Newcastle ST5 5BG, UK

[2] Department of Marketing, Hanken School of Economics, P.O. Box 479, 00100 Helsinki, Finland; bor@hanken.fi

[*] Correspondence: s.a.cropper@keele.ac.uk; Tel.: +44-7901-760256

**Abstract:** In their treatise on meta-organization, Ahrne and Brunsson theorize a distinctive organizational form, the association of organizations. Meta-organizations have the properties of formal organizations—boundaries set by determinations of membership, goals, a centre of authority, and ways of monitoring and sanctioning member behaviors. The theory draws a strong distinction between meta-organizations and networks, suggesting that similarity among members is the primary characteristic of meta-organizations, whereas networks signify complementarity and difference. Meta-organizations serve and are governed by their members, though the meta-organization itself may develop its own agency and may regulate its members. It is on this basis that Ahrne and Brunsson develop an account of the dynamics of meta-organizations, placing less emphasis on external sources of change than on the internal relationships between members and the meta-organization itself. This paper appraises the theory of meta-organizations, using a case study of Partners in Paediatrics, a subscription association of health care organizations, as the empirical reference point. Data about this partnership's membership and its activities are drawn from 12 'annual reports' covering a 17-year period. Focusing, particularly, on the membership composition of the Partnership and its relationship to the changing environment, the case analysis traces the changing character and circumstances of the Partnership, identifying four distinct phases, and raising questions for meta-organization theory and its account of meta-organization dynamics.

**Keywords:** dynamics; boundaries; change; co-evolution; meta-organization; partnership; institutional environment; composition; membership; healthcare

## 1. Introduction

There is a growing literature on the dynamics of cooperation among organizations (see Cropper and Palmer 2008), though Bell, den Ouden, and Ziggers (Bell et al. 2006) argue that the field is "*fragmented, lacks coherence, and has produced non-comparable research*". In their review of 22 longitudinal cases of inter-organizational collaborations, Majchrzak et al. (2015) start to draw together insights into dynamics, conceived as changes in the characteristics of inter-organizational collaborations and as patterns of relationships between sources and characteristics of inter-organizational collaborations. Most studies of dynamics focus on alliances between two firms, often taking the perspective of one side of the alliance (Bell et al. 2006). In this paper, by comparison, we focus on associations composed of multiple organizations, drawing on, and appraising, aspects of the theory of meta-organization (Ahrne and Brunsson 2005, 2008). Majchrzak et al. (2015) include only one study on the dynamics of meta-organizations, the study of Sematech by Browning et al. (1995), and no specific attention is given to the difference in dynamics. The (relatively limited) literature on organizational federations and associations does contain some studies of importance (e.g., Lowndes and Skelcher 1998; Selsky 1998;

Traxler 2002). However, insights into the specific character of meta-organizations remain limited and fragmentary (Berkowitz and Bor 2018; Berkowitz and Dumez 2016).

This paper presents changes in the membership composition of a meta-organization over a 17-year period and discusses the significance of these changes for the character of the meta-organization. In the case study, we draw on the annual reports of Partners in Paediatrics (PiP), from its initiation in 1997 until 2014, to capture what we term the 'compositional dynamics' of this formalized association of organizations (Cropper 2001), which meta-organization theory suggests will have distinctive characteristics. We understand 'character' in institutional terms. As Selznick (1957) observed, "*This patterning [character] is historical, in that it reflects the specific experiences of the particular organization; it is functional in that it aids the organization to adapt itself to its internal and external social environment; and it is dynamic, in that it generates new and active forces, especially internal interest-groups . . . .the emergence of organizational character reflects the irreversible element in experience and choice*". In this paper, we ask whether closer attention to the institutional environment, to change over time, and to changes in membership composition would both indicate and explain changes to meta-organizational character and strengthen the core claims of the theory of meta-organization.

The paper is organized as follows. We start by discussing the characteristics of meta-organizations and review the literature on dynamics relevant to this associational form of organization. We describe our methodology and the data included in the study. This is followed by an account and discussion of the dynamics of the case against the terms of meta-organization theory.

## 2. Characteristics of Meta-Organizations and Key Sites of Dynamics

In a recent series of publications, Ahrne and Brunsson (2005, 2008, 2011) have proposed that meta-organizations are a distinct type of inter-organizational entity (Cropper et al. 2008). They stipulate some characteristics of the form: "*meta-organizations are all associations; membership is voluntary and members can withdraw at will. The purpose of a meta-organization is to work in the interests of all its members, with all members being equally valuable and membership being based on some form of similarity*" (Ahrne and Brunsson 2008, p. 11). Although, in these senses, meta-organizations may be likened to other associations, Ahrne and Brunsson (2008) emphasize the significance of the composition of meta-organizations: Members are organizations not individuals.

Ahrne and Brunsson (2008) also stipulate a formality of character. A meta-organization "*is not the same as a network, class, or society. For people to believe that something is an organization, it must have members, a hierarchy, autonomy, and a constitution.*" (Ahrne and Brunsson 2008, p. 45). Such formalization, seen as a process and an outcome (Vlaar et al. 2006), creates the conditions for a 'decided order' (Ahrne and Brunsson 2011; Ahrne et al. 2016). Decisions concerning members specify who are in, and who are out and so set the boundary. The hierarchy exists as a center of authority: Though this may be no more than a mechanism for making decisions, it may equally hold the right to issue commands and rules prescribing members' actions. Such rights and rules are laid down in constitutions, which may also describe the goals or tasks of the organization. Ahrne and Brunsson (2008) observe that a meta-organization can gain a degree of autonomy, and become recognized as an organization in their own right. Formalization can have positive effects in terms of a meta-organization's presence and agency; however, it may also provide false impressions of comprehensibility and controllability (Vlaar et al. 2006).

In their account of dynamics, Ahrne and Brunsson (2008) focus substantially on the composition and order of the meta-organization, arguing that "*meta-organizations can best be understood as being in a transitional phase between a weak organization with strong members and a strong organization with weak members*" (p. 132). Like other accounts that focus on inherent instabilities (Das and Teng 2000), tensions (Huxham and Vangen 2000), or dialectics (De Rond and Bouchikhi 2004) in inter-organizational entities, meta-organization dynamics are held to arise from the interplay between members, the meta-organization, and their collective activities. Although Ahrne and Brunsson (2008) explain the formation of a meta-organization as a move on the environment, they say less about the way the

meta-organization might continue to respond to field influences as a source of change. The boundary between organizations and their environment is moved, but a new boundary and set of relationships both within the meta-organization and between the meta-organization and its environment are produced. Other accounts of organizational change, including the co-evolutionary perspective, recently reexamined by Rodrigues and Child (2003, 2008), frame dynamics essentially in terms of the mutual and reciprocal influence between an organization as a collective effort or strategy and its environment (see also Barnett et al. 2000; Selsky 1998). We move on to discuss institutional environment and membership composition as sources of dynamics using a third lens—that of patterned transitions through time, i.e., history.

### 2.1. Institutional Environment

The environment of organizations is unpredictable and meta-organizations are an attempt at organizing salient parts of the environment (Ahrne and Brunsson 2008). As Ahrne and Brunsson (2008, p. 56) note, "*Creating meta-organizations entails the reduction of environment and an increase in organization—transforming part of what was once the members' environment into organization. Instead of constituting each other's environment, the organization's members become members in the same organization.*" A new and additional organization is formed while no organization is 'lost' (unlike in a merger or acquisition). Members, thus, "*retain their organizational boundaries, but a new boundary is placed around them all.*" (p. 64). While the meta-organization organizes part of the environment, the environment does not disappear. Ahrne and Brunsson note, "*To a great extent, organizations must accept the environment as it is and adapt to it rather than attempt to control it.*" (Ahrne and Brunsson 2008, p. 56). However, the meta-organization still needs to deal with its environment and so do the members. Frequently, meta-organizations seek deliberately to influence and to change their environments (and that of the members) for the benefit of the members. Meta-organization theory says little about this ongoing work. In his analysis of the developmental dynamics in non-profit federations, Selsky (1998) observes that the federation's dynamics include both forms of adaptation and efforts at construction. Adaptation consists in the federated referent organization's (FRO; Selsky's term is conceptually close to meta-organizations) responses to "*influences and pressures from constituent member organizations, domain elites, and the wider context of resource and policy environments in which they operate.*" (p. 298). Construction is where the FRO seeks to shape the context through its own presence and actions. This can be understood in terms of co-evolution (Rodrigues and Child 2003), whereby the environment influences the meta-organization, but, at the same time, the meta-organization influences the environment. Rodrigues and Child (2003) recognize three system levels, that of the macro (the general environment), the meso (the immediate environment), and the micro (the organization internal), which interact. Preempting Rodrigues and Child (2008) work, Selsky concludes that the developmental dynamics of the FRO are characterized by a continuous interplay between the FRO and its context, in which it seeks to maintain effective alignment between its strategy and the significant elements of the field within which it is set. However, becoming an actor that can influence its environment requires a high degree of coordination and a strengthened common identity, according to Ahrne and Brunsson, and may increase the need for similarity among members. In addition, one of the issues with meta-organizations is that the member organizations as individual organizations also remain embedded within their environment; they can act and attempt to influence their environment as well. We expect much more of an active and complex relationship to the remaining environment than simple acceptance.

### 2.2. Membership Composition

Membership of a meta-organization is a voluntary decision by organizations: "*As members, they keep most of their autonomy and identity as independent organizations*" (Ahrne and Brunsson 2008, p. 3). Organizations need a reason for becoming and remaining a member, and many organizations may choose to stay outside meta-organizations they would be eligible to join. This is particularly the case for those with the ability to influence their environment successfully on their own, or where

they expect status, identity, or operating flexibility to be maintained outside the meta-organization. Staying outside a meta-organization can, however, "*reduce the possibility of interacting with its members or worsen the conditions for interaction*" (Ahrne and Brunsson 2008, pp. 87–88) and it can raise questions and speculations from the environment about why it is not a member.

Ahrne and Brunsson (2008) assume that the decision by organizations to become a member and the decision to accept members is based on some form of similarity, belonging to a group of organizations. Yet they also note "*many of the conflicts that exist in meta-organizations are about the extent to which the members should be similar to each other*" (Ahrne and Brunsson 2008, p. 100). A lack of similarity may lead to difficulties as it may make it difficult to identify meaningful interaction or collective action for all members. They suggest that meta-organizations can create a different category of members, associate members, to allow non-similar members to engage, while preserving the similarity among main members. The nature of the required similarity among members remains, however, somewhat unclear. Their specification is that members "*perform similar tasks and strive for similar things*" (Ahrne and Brunsson 2008, p. 60). Berkowitz and Dumez (2016) found that the membership of oil and gas organizations in meta-organizations shows significant variety with membership commonly from different sectors and hence different types of organizations (public, private, non-governmental organizations, etc.). Similarity, may then be found more in identification with the purpose of the meta-organization than in performing similar tasks or, as Ahrne and Brunsson suggest, some other 'family resemblance'. The focus of Ahrne and Brunsson, as well as other scholars writing about meta-organizations, has primarily been on the relation between the members and the meta-organization. Because they stipulate the criterion of similarity so strongly, they say less about the composition, about the limits to variety or difference, or the ways of handling difference and change in composition over time.

In what follows, we consider the dynamics that arise at the external boundary—those between the meta-organization and the environment—and those generated by the composition of the meta-organization as seen in terms of its membership and the activity that is decided and pursued on their behalf by the meta-organization.

## 3. Method & Data

Using a single-case study design, we explore aspects of a significant phenomenon under rare circumstances (Eisenhardt and Graebner 2007), a partnership's life from its inception in 1997 to its position in the healthcare sector in the UK 17 years later. We draw on the series of publicly available reports, published annually or biennially (they are referred to as Annual Reports or Partnership Reports) that offer narrative accounts of the work of 'Partners in Paediatrics' (the Partnership). The twelve reports published during this period give a rich, cumulative, and reflective account of the purpose, membership, and activities of the Partnership. Typically, the reports profile the activity of the Partnership during the reporting period and discuss its significance in relation to the Partnership's purpose and context. The Reports do not give a clear sense of the relationship between the meta-organization itself and its members, except in the discussions of what has been achieved or not achieved, by the Partnership on behalf of the membership. They say little about members' motives in joining or leaving the Partnership. Nevertheless, they offer a sense of pattern through time. We have summarized the Reports, year-by-year and section-by-section, tracking the changes in Partnership structure, membership, work streams/activity, resourcing, and claims about achievement and impact. There are clear continuities in each of these, but there is also disjuncture. As well as exploring the internal bases of change, we have also noted where the Reports comment on how it is connected to its environment, both through the member organizations, but also directly as an actor in its own right. Duriau et al. (2007) note the value of annual reports as a source of data for longitudinal analysis, they also caution that there are limitations, e.g., such reports tend to be strongly biased to positive accounts of the organization, and they present, essentially, a singular view. We recognize this and the potential for a lack of chronological precision given the irregular production of the reports and the movement between the Partnership age and calendar date across the reports. Nevertheless,

the content of the reports allows for checks on sequence, period, and date, and the tone of these twelve reports is not wholly self-satisfied: As much as reporting success, they identify and explain tensions and frustrations, not least in the Partnership's capacity to produce change. Several of the reports take 'change' as the central theme and offer reflective comments in which connections are made between the Partnership, its ambitions and capacities, the receptiveness, or otherwise of the Partnership's environment, and the degree of alignment between these.

Analysis of this data allows us to explore both the question of the character of the Partnership, in terms of the formal characteristics of meta-organizations, and to track certain forms of dynamics that have been significant in the evolution of the Partnership. This will also allow us to offer an empirically informed appraisal of important arguments within meta-organization theory.

We start with a brief account of four phases in the life of the Partnership:

(i)    Initiation and formalization of the Partnership;
(ii)   a period focused on the promotion and organized exploration of 'wide area managed clinical networks' as a means of organizing pediatric services;
(iii)  a stalling of energy and progress on wide area managed clinical networks; and
(iv)   competition for wide area network leadership, and a split agenda within the Partnership.

These phases are a temporal bracketing (Langley 2010) in which we identify continuities within phases and discontinuities in character at the frontiers. Between phase (i) and (ii), the discontinuity concerns changes in the constitution of the Partnership, and, particularly, its formalization. Between phase (ii) and (iii), the discontinuity concerns the change in membership composition and a stable programme of collectively organized activity. In addition, between phase (iii) and (iv), the discontinuity concerns institutional pressures (conducive or adverse/disrupted policy and influence from powerful actors, both within the Partnership and from outside). This next section serves to introduce the partnership, its evolution through 17 years, and events that have affected its character. The section that then follows comments specifically on the compositional dynamics of the Partnership, a consequence of internal tensions that arise from the meta-organizational form, but also from the circumstances that shape its membership, and from the patterns of membership themselves. The paper concludes with a brief response to Ahrne and Brunsson (2008) specification of the character of meta-organizations, and some suggestions for future research.

## 4. *Partners in Paediatrics*: Environment and Partnership (Co-)Evolution

In this section, we draw out two sets of observations about Partners in Paediatrics (PiP) as it has evolved over time and about the sources of this change. First, we comment on the effects of the changing environment on the Partnership. Like recent studies of co-evolution (Rodrigues and Child 2003, 2008), the UK health sector—the National Health Service (NHS)—is rich with institutions, and is highly politicised. The professions have influence and the administrative hierarchy remains a significant factor in institutional and ideological forces, which are seen as the process of translating the most significant of influences, policy—or rather myriad policies, often ambiguous, and contradictory—into effect. We also map the changing composition of the Partnership, and the effects that a changing membership mix has on the character of the Partnership.

*4.1. Partnership (Co-)Evolution: Four Phases of the Meta-Organisation in Its (Changing) Environment*

4.1.1. Phase 1: Initiation, Formation, and Formalization

Ahrne and Brunsson (2008, p.   79) observe that the meta-organizations they studied "*typically started at conferences to which all relevant parties were invited*." The Partnership was, indeed, initiated by representatives (pediatricians and managers) of nine general hospitals (NHS provider organizations are constitutionally designated as 'Trusts') who met in November 1997 to "*discuss the opportunities for developing greater collaborative links between providers of paediatrics in this geographical*

*area . . . There are already signs that collaborations are developing in an informal way . . . "* . . . . (Letter of Invitation, 6 October 1997).

Behind the invitation and the list of invitees was concern about the effects of the internal market established within the NHS from 1991 on the planning and resourcing of pediatric hospital services. The internal market separated provider organizations from commissioners of services, the latter being charged with defining and securing services to meet the health care needs of their local population. Commissioners hold authority over the organization of services and access to services through their decisions about the services they specify and fund. As Ahrne and Brunsson (2008, p. 65) observe, *"If a field already has a high degree of order, one might think the need for a meta-organization is less; yet another type of order may be desired*." The call for greater collaboration was in the face of incentives towards competition between provider organizations, a lack of meaningful dialogue between commissioners and provider organizations about services required, especially more specialized services, and how best to secure them. At a second meeting of this group one month later, the group agreed on a Statement of Purpose, outlining the rationale and proposed activity of a Partnership:

> *"The driving purpose of the collaboration is to improve the quality and accessibility of services for children across the area served by the participating hospitals.".* (AR1: p. 3)

Three objectives were also specified:

- Balancing local needs and provision of high quality specialist services;
- managing manpower, training, and research; and
- Advising commissioning agencies and groups re. paediatric services. (AR1: p. 3)

The three objectives can be seen consistently to have guided the activities of the Partnership.

The initiating group of hospitals could be said to have two similarities—identity and interest. In terms of similarity of identity, the nine hospitals invited to the first exploratory meetings were all general hospitals providing secondary care services for their local populations (250–400,000): All had pediatric divisions, generally small compared to other divisions, e.g., medical or surgical, within the hospital. In addition, all served communities in the triangle set between three major cities and their Children's Hospitals. The Children's Hospitals provide specialist, tertiary services to the wider region and its population (usually several million), and potentially very specialist services for still larger populations. However, they also provide general secondary services for the children within their city hinterlands. In part, because of difficulties of access to the tertiary services in the Children's Hospitals, but also because Pediatrics was following the tendency within medicine towards greater specialization of training and practice, the larger general hospitals were starting to develop specialist services within their general pediatric offer. Nearby, smaller, general hospital units were then referring patients to them as well as, or instead of, to Children's Hospitals for specialist opinions and treatment. Among the nine general hospitals that formed the Partnership, there were shared interests and concerns about a) the quality and range of healthcare provision in the general hospitals, even the larger ones, could, themselves, viably provide, and b) whether access to specialist services that the Children's Hospitals were funded to provide was fair or adequate. They agreed it was not.

From the discussions and the statement of the purpose that was fashioned by this group came a decision, relatively rapidly, to formalize arrangements. The first annual report notes:

> *"The group is suggesting strengthening the organizational basis . . . as a means of identifying and pursuing shared agendas where joint action could bring benefit. This would also mean some formalisation of the Partnership itself: its boundaries, so far open, might be more visible and the terms of membership more defined and conditional on appropriate contribution to the work of the group."* (AR1: p. 13)

By September 1998, as the Partnership moved towards a more formal constitution, representatives of 14 Trusts were actively involved. At the [second] Conference, there was a report:



*"to the Chief Executives of the participating Trusts on the progress made to date and to seek a mandate to continue with this work . . . . the Chief Executives requested a written summary report before agreeing to a financial contribution to support the further work of the Partnership . . . . A further 'Open' meeting was held in January 1999, to conclude the work of Year 1, to agree the programme of work for Year 2, establish the formation of the Steering Group and appoint its 'officers' . . . "* (AR1: pp. 1–2)

The agreement of the Chief Executives signaled, it was understood, the commitment of each member organization to the Partnership. Subscription fee levels varied depending on size of hospital, but, in all other respects, it was a 'partnership of equals' represented on the Partnership Steering Group. (AR1, p. 5). There is a reminder of the governance arrangements in each Annual Report and many more detailed explanations in the early years (AR1, pp. 4, 12–19; AR2, p. 7; AR3, pp. 5–6) and in later Reports from the Chair of the Partnership (AR7, p. 3; AR12, pp. 1–2).

Not all Trusts involved in the formation joined: Four community health trusts, responsible for providing health care outside hospitals to local communities, did not continue into Year 2. However, another community health trust did join the general hospitals in founding the Partnership, and there is evidence in an evaluation reported in AR3 that the issue was one of the relevance of the agenda and priorities, which focused on secondary and tertiary care rather than the community- and home-based care that was the main business of this other type of health care provider organization.

Appointed officers of the Partnership formed a core group, a management and coordinating center for the Partnership. Ahrne and Brunsson (2008) stipulate a 'center of authority', responsible for decision-making within the meta-organization, which we take, in this instance, to refer to the Steering Group: They also note that the meta-organization may be allowed, or may develop, a certain degree of agency independent of the members, and that the dynamics of governance between members and the meta-organization itself are a matter of central interest in understanding the character of any meta-organization. Early in the Partnership's development, the sense of collective and mutual responsibility was promoted. For example, a coordinated action to map the availability of specialist practitioners in the member Trusts reported that:

*"The Partnership members have all agreed to adopt a protocol to share information and accept influence from the Partnership concerning the appointment of consultants and this, together with the development of a strategy and service plan, will be the foundation stones for provider collaboration in service delivery in the future."* (AR2, p. 13)

Attention to governance, as a means of deciding on its programme of activity, as a means of rendering account, and as a source of legitimacy with members (Human and Provan 2000), reveals the careful development of the (internal) relationship between the Partnership and its members. However, it is equally clear from the Reports that the Partnership was also developing its connections to the wider scene.

### 4.1.2. Phase 2: Wide Area Managed Clinical Networks

The formalization of the Partnership coincided almost with a significant change in the policy environment. First, a trenchant critique of policy attention to children and young people (Aynsley-Green et al. 2000) meant that the government was urgently issuing new national guidance. The first tranche (Department of Health 2003) focused attention on the quality of care within the hospital setting. AR5 (pp. 5–6), issued at the end of 2003, provides a detailed comparison of the policy mandate, with the work of the Partnership boosting the legitimacy of the Partnership, and strengthening the hand of those stakeholders who would recognize the Partnership's value. The policy also asks for closer partnership among organizations. In this second phase of the Partnership, an idea becomes central to the work of the Partnership. A 'managed clinical network' is defined as *"linked groups of health professionals and organisations from primary, secondary and tertiary care working in a co-ordinated manner, unconstrained by existing professional and existing [organisational] boundaries to*

*ensure equitable provision of high quality services*." ([Baker and Lorimer 2000](#), p. 1152) This idea captures exactly the task set by the Partnership. It was also starting to receive significant attention nationally (e.g., [NHS Confederation 2001](#)). In both respects, the Partnership was ahead of the crowd.

The (long) second phase of development of the Partnership involved a series of projects, several running at any time, to apply the idea of the managed clinical network to the services identified as priorities for improvement. This work evidently took the Partnership beyond its membership. Thus, we observe a distinction drawn between the members of the Partnership (listed in each Report) and the full range of participants that are reported to be contributing. Members have attendance and voting rights at the Steering Group/Board of the Partnership. Moreover, they are mentioned as privileged beneficiaries of the Partnership. However, participants are also important. The following appears in a number of the Annual Reports:

> *"We are an open organization and as such are keen to work with anyone to help further our aims. Please get in touch if you are interested in getting involved in any of our projects."*. (AR4, p. 2)

Participants, from non-member organizations help to populate the Partnership's Working Groups. They bring reputation, expertise, and energy, and act as ambassadors to their own organizations and to other significant stakeholders. This cast of agents at, or just beyond, the boundary of the Partnership, but who also are invited into, or encroach on, the Partnership, has clear significance in making and sustaining the Partnership. The Partnership's boundaries are in place, but they are low and porous, and the presence and actions of non-members shape the character of the Partnership.

Within this period, there was also significant interchange between the Partnership and other stakeholders. Learning is shared widely among a range of beneficiaries rather than restricted to membership as 'club goods' would be. At the macro level, the Partnership was recognized nationally, and internationally, for its work in exploring how managed networks could work for pediatric services, even if market practices and a strong sense of hierarchical control remained central to the NHS. At the meso-level, the Reports suggest that its most important external relations are at the regional or strategic levels, where the scale of the geographical responsibility of NHS organizations matched that of the Partnership and its focus on wide-area managed clinical networks for specialist services. There are references to strong links to the Strategic Health Authority and to projects supported and funded by that influential external organization. The Annual Reports start, after five years, to suggest that the more localized, commissioning organizations that purchase health services from providers for their populations are also engaging with the Partnership: "*Commissioners are well-represented as advisers . . .* " (AR5, p. 5), and "*PiP has a lot of knowledge about services that could assist the commissioners . . .* " (p. 6). AR6 then reports that the Partnership has been able to establish a means of conducting relations with the commissioners, at a scale that would potentially enable coordinated action: "*This year, PiP and commissioners have formed the Paediatric Specialised Services Steering Group. The group's overall objective is to develop strategic planning advice with commissioners for development and deployment of paediatric services across . . . .the PiP area. This is a huge step in taking PiP's service projects forward*" (AR6, p. 3).

### 4.1.3. Phase 3: Wide Area Networks Stalling

However, ARs 5 and 6 also suggest a turning point, in part as a consequence of continuing changes to the organizational structure of the NHS. AR5 notes a 'whole new set of challenges': "*It had always been an issue for PiP how most effectively to engage with multiple commissioners. . . . With the advent . . . .of Primary Care Trusts (PCT), PiP now has a significantly altered landscape within which to plan and develop services. On the one hand, the environment is more complex; fourteen PCTs as opposed to five Health Authorities . . . On the other hand, PCTs are not solely commissioners but also significant providers of services to children. This latter point represents an opportunity for PiP*." (AR5, p. 3). The membership now includes three PCTs, which have both commissioner and provider responsibilities—the internal market structure had been compromised by the latest NHS reorganisation. The commissioning arm was responsible for securing all children's secondary care services and access to tertiary care. That these organizations are

members is therefore significant. We note, however, that it is likely that PCTs became members because community services for children have been transferred from what were member organizations, now reorganized, into their newly-mandated provider 'arms'.

The Partnership also notes that "*…ultimately PiP is only as strong as its member commitments will allow.*" (AR5, p8) and, most tellingly: "*Where members have been most frustrated is in implementing proposals for collaborative service change and development. As a partnership, PiP has no power or authority to effect change: authority lies with individual members and through the commissioning systems. Where issues require a view across services, or across health localities, there has often been uncertainty about whose responsibility it is to mandate, lead, or take action. PiP has been able to help in coordinating action where there is commitment from interested parties.*" (AR6, p. 6). Subsequent Annual Reports, therefore, reflect on the limits to the Partnership's ability to effect planned change. Initial steps to implement plans for networks in pediatric gastroenterology and in surgery are reported, with investments by groups of members on behalf of the Partnership. However, these do not pay off as planned, in part, because supporting investments from other members, needed to develop and sustain the new planned patterns of service delivery, do not materialize in time. The Partnership is not alone in this, but it has evidently understood that Baker and Lorimer's (2000) warning that managed clinical networks are difficult to implement and sustain is well-founded. Second, the serial reorganization of the local NHS not only affects the membership composition of the Partnership, but also disrupts its immediate environment, with repeated change to commissioning organizations and to the regional/strategic tier at which scale the managed clinical networks would have had a natural alignment. (see Paton 2014).

### 4.1.4. Phase 4: Split Agenda and Competition for Field (Meta-) Organization

Column 5 of Table 1, as shown below, shows that membership of the Partnership had incrementally become more varied. The general hospital trusts were no longer the dominant majority: A substantial minority of members are community service trusts, and yet the activities of the Partnership have only marginally met the interests of these organizations. AR5 explores different future programmes of activity: One of these involved "*drawing away from PiP's focus on care for the acutely ill child.*" Then, AR6 reports that a project had been initiated to address issues in Child Protection. As it had done for the specialist acute services, the Partnership had seen, earlier than other agencies, problems with the quality and accessibility of services. It assertively pursued this problem. A project group both incorporated and supported Police and Social Services organizations and health service commissioners. Its work was covered in each subsequent Annual Report, including, ultimately, a note reporting tangible improvements to services.

By 2009, a split in activity and in the Partnership is apparent, reflecting different member interests and a changing relationship between the Partnership and one powerful member, the Children's Hospital, responsible for tertiary services to the general hospitals that are still a majority of the members. It is the only Children's Hospital that remains a subscribing member. Ahrne and Brunsson (2008) note that when a meta-organization and its members become too similar, there is no need for the meta-organization. AR10 reports that the Children's Hospital "*approached PiP with a proposal to use their membership to work collaboratively in developing regional specialist networks. They plan to use the independence of the PiP Board to ensure that developments are prioritised and owned by providers … .and that the requirements of all members are accounted for at the planning stage. In order to take this on the main change for PiP will be the requirement to accept new members from across the region. PiP members are currently considering this proposal.*" (AR10, p. 2). AR11 reports "*we have continued to work more closely with [the Children's Hospital] in the work around developing a strategy for specialist clinical networks*" and, significantly, "*establishing clear governance arrangements … will be achieved through creating a Board and two sub-committees; one focusing on acute service networks and the other focusing on integrated community pathways.*" (AR11, p. 3). This 'internal jolt'—the division of activity and of member interests into wide area networks and local service projects—was reported as proceeding. However, AR12 also reports two external 'jolts'. The new government, elected in 2010, passed the Health and Social Care Act 2012. Despite promises not to

pursue more structural change, it does so, creating still more fragmentary and numerous commissioning organizations—now called Clinical Commissioning Groups (CCGs). AR12 also notes the '*establishment of the [regional] Strategic Clinical Network [SCN] (Maternity and Children) and the Clinical Senate.*" Whilst an official endorsement of the clinical network as a core mode of organizing by the NHS, the report of the annual conference for 2012 notes challenges to the Partnership's role, stating that "*We need to bring existing local/informal networks into the work of the SCN . . . and involve all stakeholders.*" For a Partnership that had been recognized for the depth and richness of clinical engagement, and for its work to develop and support informal and formal clinical networks, this was a significant threat. Nevertheless, AR12 remains optimistic:

> "*PiP has continued to be a stable and influencing advocate for children and young people. The focus on improving health outcomes . . . across the region remains as strong as it did when the organization started in 1997. The organization has actively responded to the 'external' challenges—whilst also reviewing itself as an organization and strengthening its governance arrangements.*" (AR12, Chair's Foreword, p. 2)

The focus for the future (AR12, p. 13) sets out activities that are less about change through the redesign of services as networks—representation of members, facilitation of interaction and support to informal networks, education and sharing good practice, developing and monitoring standards, guidelines and quality improvement measures, and engagement with children and young people as service users. These match more closely to the functions of a meta-organization discussed by Ahrne and Brunsson (2008). In terms of the Partnership's membership, there is a broadening of the appeal: The Report advises "*All NHS and social care organizations involved in, or concerned with the provision, commissioning or regulation of services for children and young people are eligible to become a member.*" (AR12, p. 15). Yet, the membership profile shows a clear return to the general hospital as the dominant member type.

Table 1, as shown below, provides a more systematic enumeration of the membership of the Partnership, and change in membership composition year by year. It also notes the degree to which the policy environment is conducive to attention and action on children's services and to collaborative approaches to service provision and improvement. Both affect the Partnership's ability to attract membership and to influence, through work with external organizations, the environment to the members' advantage.

**Table 1.** Membership of the Partnership: Composition and changes 1997–2014.

| Partnership Phase | Date | Annual Report | Membership | | | Conducive Environment | |
|---|---|---|---|---|---|---|---|
| | | | Number of Members | Similarity | Change/Continuity and Composition | Policy on Collaboration & Networks | Policy Emphasis on Children |
| *Initiation and formalization* | 1997 | AR1 | 9 | Very high | 9 general hospitals | No | No |
| | 1998 | AR1 | 11 | High | **2 new**: 2 community service trusts<br>→ 9 general; 2 community | No | No |
| | 1999 | AR2 | 11 | High | No change | No | No |
| | 2000 | AR2/3 | 13 | High | **2 new**: 1 general hospital; 1 community services provider<br>→ 10 general; 3 community | No | No |
| *Wide Area Managed Clinical Networks* | 2001 | AR3/4 | 18 | Varied | **5 new**: 3 specialized Children's Hospitals, 1 community services trust, 1 Primary Care Trust (PCT)<br>→ 10 general; 4 community, 3 specialized, 1 PCT | Supportive within market | Very positive |
| | 2002 | AR5 | 19 | Varied | **5 new**: 4 PCTs; 1 community services trust<br>**4 loss**: 3 community services trusts assimilated into PCTs; 1 PCT<br>→ 10 general; 2 community; 3 specialized, 4 PCT | Supportive, within market | Yes |
| | 2003 | AR6 | 19 | Varied | No change | Supportive within market | Yes |
| *Wide Area Networks Stalling* | 2004 | AR6 | 17 | Varied | **2 loss**: 1 specialized Children's Hospital (out of main region); 1 general hospital (2 members merge)<br>→ 9 general; 2 community, 2 specialized, 4 PCT | Disturbed | Yes |
| | 2005 | AR7 | 17 | Varied | No change | Disturbed | Yes |
| | 2006 | AR7 | 17 | Varied | No change | Disturbed | Yes |
| | 2007 | AR8 | 15 | Varied | **2 loss**: 1 general hospital, 1 community services trust<br>→ 8 general; 1 community, 2 specialized, 4 PCT | Yes | Yes |
| | 2008 | AR8/AR9 | 16 | Varied | **2 new**: 2 PCT<br>**1 loss**: 1 PCT (out of main region)<br>→ 8 general; 1 community, 2 specialized, 5 PCT | Yes | Yes |

**Table 1.** *Cont.*

| Partnership Phase | Date | Annual Report | Membership | | | Conducive Environment | |
|---|---|---|---|---|---|---|---|
| | | | Number of Members | Similarity | Change/Continuity and Composition | Policy on Collaboration & Networks | Policy Emphasis on Children |
| *Split Agenda and Competition for Field (Meta-) Organization* | 2009 | AR9 | 16 | Varied | No change | Yes | Yes |
| | 2010 | AR10 | 17 | Varied | **1 new**: 1 PCT<br>➜ 8 general; 1 community, 2 specialized, 6 PCT | Disturbed | Unknown |
| | 2011 | AR11 | 16 | Varied | **1 loss**: 1 specialized Children's Hospital (out of main region)<br>➜ 8 general; 1 community, 1 specialized, 6 PCT | Yes | No |
| | 2012–2014 | AR12 | 18 | High | Geographical extension of Partnership boundary.<br>**8 new**: 5 general; 2 community; 1 new style commissioner—Clinical Commissioning Group (CCG)<br>**6 loss**: 6 PCTs (abolition of PCTs)<br>➜ 13 general; 3 community; 1 specialized; 1 CCG | Yes | Yes |

### 4.2. Membership and the Consequences of Compositional Change

In this section, we consider how the composition of the Partnership, set out in Table 1, has changed, why it may have done so, and with what consequences. We focus here on membership composition and change, and, particularly, on the criterion of 'member similarity', since this is the principal point of reference in meta-organization theory.

First, we note that the Partnership has survived. The Partnership is 'of', but not 'in', the NHS. Members are exclusively drawn from the NHS and the composition of the Partnership has been heavily influenced by the embeddedness of its membership in that structure, which dominates the field of children's health care. However, the Partnership itself is not subject directly to the policy decisions that shape it and its local organizational structure. It has survived because its overall membership has been sustained at or above a critical mass.

Second, however, the Partnership's membership has changed. The question of similarity and variety in membership is not addressed directly in the Annual Reports, although there are a number of points we can infer. Membership grew rapidly in the three years following the Partnership's inception and then stabilized. It is not a large membership (19 members are listed at the maximum), and so the admission of a new partner or the loss of even one member is of some significance—these events are indicated in the Annual Reports. There has been a high degree of continuity of membership of the founder members—a group of general hospitals and one of the Children's Hospitals has remained a member since it joined in 2001 as has one Primary Care Trust (PCT), also joining in 2001. This year is the point at which the criterion of 'similarity', that Ahrne and Brunsson (2008) suggest is characteristic of the meta-organization, becomes a question of significance. Whilst these two new members may share an interest in 'improving the quality and accessibility of children's services', (taken from the Partnership's statement of purpose, this being the declared criterion by which to judge an organization's suitability to join), they are quite different types of organization.

In its first phase, the membership of the Partnership is at its most coherent or homogeneous, dominated by general hospitals concerned to improve access to specialized services, potentially by developing their own, collaboratively. The Annual Reports use a self-referential term, "*in the geographical area covered by the participating organisations*", but there are also references in later Reports to an extension of the geographical field to cover fully a region served by the long-standing Children's Hospital member. The majority of new members listed in the final Annual Report reviewed (AR12) are general hospitals, these identifying with the Partnership and with the Children's Hospital, and admitted following a decision by the Partnership to extend its geographical area to match the service catchment of the Children's Hospital. This, and the loss of the PCTs as members (abolished in a further NHS reorganization), shifts the balance of membership decisively back towards the general hospitals category.

Between these two periods, Table 1 indicates how variety has increased and how the membership has become spread, indeed split, between members of several types: First, the admission of the community service trusts, then, the organizations of primary interest to the Partnership founders, the Children's Hospitals; and the commissioning organizations also identified as essential in the objectives of the Partnership. In its most inclusive statement of eligibility to join, AR 11 (p3) has, within the Partnership's prospectus, a further broadening of intent: " . . . *not only in continuing to support the current network of health care staff across the [region] but also developing new networks across the healthcare and well-being sectors.*" The final appeal in AR12 is to the widest set of organizations interested in children's health and social care to join the partnership. The criterion of similarity, spelt out in membership decisions, has, at this moment, significantly broadened, although there is no evidence in the Annual Reports of this in the new members admitted to the Partnership—rather, the converse.

Ahrne and Brunsson (2008) note that meta-organizations may define their membership in well-established terms or establish new categories of organization to influence the organizational field. In the case of the Partnership, the criterion for membership cuts across established ways of

distinguishing NHS organizations and joins them in terms of their interest and responsibility for the quality of children's' services.

The (changing) mix of members is a significant factor in understanding the character of this meta-organization. The trajectory, or series of transitions in composition and activity, moved the Partnership to a point where there was sufficient variety in member identity, and in activity, to lead to proposals for internal division. Thus, AR11 announces "*PiP looks forward to expanding its networks, and further involvement of healthcare partners, by establishing clear governance arrangements. This will be achieved through creating a Board and two sub-committees; one focusing on acute service networks and the other focusing on integrated community pathways . . . .*" (p. 3). It is also important to note, in the complexity of identity and affiliation, the changes to the geography of the Partnership. This has come gradually, through member withdrawals and new affiliations, to align more fully with the NHS 'regions'—institutional or administrative identities. In a rich institutional context, where the meta-organization is inevitably highly embedded, there will be many competing ways of defining similarity, and tensions arising, then, in the definition of membership: Leaving the primary criterion of similarity broadly-defined invites a continuing, or recurrent, negotiation of the essential character of the meta-organization.

Third, there are some questions raised by the case about the terms in which member organizations understand the implications of membership. Ahrne and Brunsson (2008) account places the greatest emphasis on the formality of the meta-organization as an organization, and on the status of admission to membership of the meta-organization: The organization is admitted as a member and that carries with it a defined set of responsibilities and rights. Ahrne and Brunsson's conception leads them to the position that meta-organizations will be able to access all the necessary 'interests and resources' of their members. The Annual Reports indicate that the commitment of the whole organization cannot simply be assumed. This is exactly Ahrne and Brunsson's point—that the meta-organization turns the environmental order into organization—it is an intervention within a field. Members' autonomy and own governance is potentially affected by membership of the meta-organization and this may be uncomfortable and even resisted. Ahrne and Brunsson (2008, p. 65) observe "*Whether or not, and to what degree, organizations that were once part of the environment are more easily influenced and more predictable will vary from case to case. . . . The world will not necessarily be an easier place in a meta-organization than in an environment; rather it will tend to be more complex in many respects. From an environment that is often cumbersome may emerge an organization that is often cumbersome . . . *". Stringer (1967) account of the multi-organization is less demanding: He conceives this arrangement as "*the union of parts of several organisations, each part being a subset of the interests of its own organisation*" (p. 107), and variability in members' responses is then a practical and empirical question rather than a matter for definition. The key point is that variability might be expected, especially where the meta-organization is both more inclusive in its membership and less powerful in terms of its capacity to direct members and to monitor and sanction member behavior. To allow for an understanding of the variation and dynamics in meta-organizations, we suggest it is necessary to consider carefully the 'internal' boundaries of the meta-organization—that is, the extent to which the meta-organization can call upon the membership of its individual constituents for the collective purpose. In the Partnership, all members have an interest in, and responsibility for, children's care. However, that interest and identity are more or less central to each organization. For the Children's Hospital, it is all. For the general hospital, community services trust, or the commissioner, it is one of many sources of identity and one of many responsibilities. Such understandings, embedded historically in the NHS and the wider care sector, cross-cut the Partnership's defining characteristic. In forming, the Partnership has sought to strengthen the identity and status of children's' services, and the potential of the Partnership to enact this purpose becomes defined and redefined, in part, in its membership.

Changes in policy can also disturb such understandings and thinking about membership. Two examples reveal the complexities of identity and the tensions between similarity and difference that Ahrne and Brunsson (2008, especially Chapter 5) suggest characterise meta-organizations. First, the Partnership, consisting of children's health service providers, had set out to engage with NHS

commissioners. Successive Annual Reports comment on the difficulties in achieving such a bridging role between the Partnership and commissioners in its immediate environment. National policy and changes to the local organization of the NHS created a new 'hybrid' organisation—the Primary Care Trust. The Foreword to Annual Report 5 made comment on the possibilities this opened:

> *"The year 2002–2003 marked the beginning of a whole new set of challenges . . . .Not least . . . the fundamental change to the commissioning environment in which we now all operate. It has always been an issue for PiP how most effectively to engage with multiple commissioners . . . . [Now] . . . PCTs are not solely commissioners but also significant providers of services to children. The latter point represents an opportunity for PiP . . . to see the PCTs not solely as commissioners but as full partners in the provision of seamless care for children. If we all begin to use this vision as a means to inform our thinking about how PiP might develop in the future, then the year 2002/2003 may come to be seen as the dawn of a new phase in the evolving story of Partners in Paediatrics."*

Table 1 shows that PCTs did indeed become members of the Partnership, though relationships between the Partnership and NHS commissioning functions continued to be pursued formally in a forum which brought together all commissioners—non-member PCTs as well as member PCTs—but only for a part of the Partnership's geographical area. This lack of 'fit' between the structures and territories of the NHS commissioning and the Partnership has been a recurrent theme in the Annual Reports.

The second example concerns the attempt to affiliate organizations, newly-created by policy, during a later phase of the Partnership. Annual Report 7 (pp. 7, 9) talks of the new structures called 'Children's Trusts' to which the Partnership should relate. There is no record of any Children's Trust joining the Partnership. A shared concern with the quality of children's services was not a sufficiently strong source of attraction. Directly related to children's services, these organisations identified with—indeed, incorporated—the formal organisations of not only the health service and public health, but also social care, education, and criminal justice. More competitors to the Partnership than likely members, these organisations were, in any event, soon dissolved by a further round of policy change. Ahrne and Brunsson (2008, p. 105) explain this by talking of 'wrong similarities', where the primary identifications do not coincide. The Partnership attaches to the NHS and to health services for children—later attempts to extend beyond this primary identification—the child protection project, invitational messages, etc.—produced participants but did not attract new members.

Fourthly, nevertheless, we note the emphasis placed in the Annual Reports on the value of participation by non-members on the character and productive work of the Partnership. Huxham and Vangen (2000) account of the ambiguities and complexities of participation in collaborative arrangements in general helps to check and enrich the formalized sense of the meta-organization set out by Ahrne and Brunsson. In this light, it may draw too strong a boundary around the meta-organization, and place too heavy an emphasis on the principle of 'decided order'. Although the Partnership has formalities, and members have certain privileges of governance, nevertheless, the boundaries are less definite than Ahrne and Brunsson suggest. We make two points here. First, as Barnett et al. (2000) argue, the growth dynamics of collective action systems—R&D consortia in their case—is where the generalist meta-organization would have an advantage over those with narrower bases—their reference point is the range of functions and activities rather than member identities, but our case narrative suggests these may be closely related. They propose 'contagion' as the mechanism by which meta-organizations would recruit new members. From the annual reports, it might be understood that the ethos of openness has attracted interest and participation rather than membership from the wider identity group—all those concerned with children's services—rather than the narrower heartland of pediatric services. This may not be 'free-riding' behavior, but rather discrimination between associational opportunities: The Partnership has used 'participation' as a source of legitimacy and as a means of influence over external stakeholders. In addition, the extension of the Partnership into its environment is a response to membership composition. The changing relationships the Partnership has with the environment arise as membership composition changes.

The salience of the Police and of Social Services as elements of the environment arose, as the growth in membership by community service organizations demanded a strand of activity that would engage those members and justify their continuing membership. Second, in their account of the 'external control' of organizations, Pfeffer and Salancik (1978) suggest that it is helpful to consider organizations not simply in terms of the 'formal boundaries' defined by membership and structure, but to consider behavior and activity as the in vivo points at which to cut the web of organization. In this sense, the analysis not just of affiliation and support, but also of activity, is of importance. The formalities of Partnership governance are given attention: AR12 starts with a statement from the Chair that "*The organization has actively responded to the 'external' challenges—whilst also reviewing itself as an organization and strengthening its governance arrangements.*" (p. 2). However, the dominant message is that the boundaries of significance to the members are set by the meaningful and productive activity of the Partnership rather than the niceties of its formal constitution. Outcomes are not wholly in the Partnership's control and there is frequent reference to the need and desirability of close connection to the health service commissioners, to the Strategic Health Authority, to Children's Trusts and the Directors of Children's Services, and so on. The external work is also 'internal work', and the Partnership is, consequently, emergent as much as decided, dependent, at the least, on decisions of others about how to manage their environment, including the Partnership.

## 5. Conclusions

There is much in Ahrne and Brunsson's account of meta-organizations that is recognizable in our case study. We note the formalized character of the Partnership and the effort made to produce a 'decided order', and the Partnership as 'an attempt' to reshape and influence the field of interest; and we note the work of the Partnership both to structure the contributions of and interactions among members, and to promote collective presence, voice, and action. The theory of meta-organization has validity as a means of asking questions of, 'seeing' and describing, analytically, the character and dynamics of the Partnership as a meta-organization. However, our analysis of the accounts of the Partnership nevertheless raises questions for the theory of meta-organization.

First, the theory of meta-organization takes a strong position on the characteristics that define the form, and Ahrne and Brunsson (2008, p. 92) suggest that where difference and complementarity are the basis for engagement an associational form would not be a natural choice; rather, they would expect a network. The case study of PiP suggests two possibilities for consideration. Either the partnership, both in its varied membership composition and in its pursuit, as a primary aim of collective action might be a hybrid, for example, a 'whole network' in Provan et al. (2007) terms, formally constituted and bounded, but with accepted variety. Alternatively, the Partnership may have evolved through phases from association to network and back towards association, in response to experience in and signals from its environment, but also seeking to follow the will of its members and to distinguish the Partnership clearly from its members' capabilities. This would be the explanation of co-evolutionary scholars. A third possibility looks to the theory: Meta-organization theory may find it important to respond to empirical studies and to other, relevant theoretical traditions. In doing so, it might 'unbound' itself from its more stringent assumptions, and tolerate a more diverse set of characteristics of the form. Berkowitz and Dumez (2016) also suggest the criterion of similarity may need further thought.

Second, since the theory of meta-organization focuses largely on internal dynamics, it makes sharp distinction between what is inside—the member organizations—and what is outside—all other organizations. Our case study, however, suggests that beyond this 'governance boundary' of formal membership, the environment remains highly salient. In Rodrigues and Child (2008) terms, co-evolution continues: Indeed, the differences between institutional environment and internal organization are often less readily discernible and are more closely intertwined than Ahrne and Brunsson suggest. Pfeffer and Salancik point out, " ... *the boundary is where the discretion of the organization to control an activity is less than the discretion of another organization or individual to control that activity*" (Pfeffer and Salancik 1978, p. 32). The Partnership is a decided order, but it is also an emergent

order. It is associational in form and practice; however, it also promotes and is understood by members as catalyzing and delivery changes to the production systems of members, and, by implication too, of non-members who form part of the system of children's services. Ahrne and Brunsson admit such 'collaboration' to the list of reasons for creating meta-organization, but they are pessimistic about its success.

This paper has examined the changing membership composition of a Partnership, which has many of the characteristics of a meta-organization as set out by Ahrne and Brunsson (2008, 2011). It traces the growth and stabilization of membership, enabling and limiting factors, and focuses on the pattern of membership composition over time—we term this 'compositional dynamics'. Ahrne and Brunsson's account stipulates an essential similarity of identity of members as a defining characteristic of a meta-organization. The paper explores what similarity might mean, how that criterion might break down, and what consequences compositional diversity might have for the meta-organization. Following the membership of the case study, Partnership, over some considerable time (17 years), we have been able to explore some effects of membership change on the focus and structure of the meta-organization, what it hopes to reach, and what it does. The paper has also suggested that the boundaries between the meta-organization and its environment and between members and non-members needs further attention in meta-organization theory. We note that the institutional environment remains of significance as a system of meanings that help to define similarity or difference between existing and potential members. Further, active non-members could be recognized as playing a potentially important role, if not in the governance, then in the energy and action radius of the meta-organization. Both questions about the boundaries of meta-organization in theory and practice would merit further case and comparative case research. The relationship between membership composition and the pattern of meta-organization activity over time is also an important area in the assessment and continuing elaboration of this novel and intriguing theory. Such detailed empirical studies will also help in appraising the arguments in theory about the distinctive character in meta-organization and its relationship to other (adjacent) types of inter-organizational entity.

**Author Contributions:** The authors contributed equally to this work.

**Funding:** The Strategic Research Council at the Academy of Finland, project CORE (313013 + 313017), made the contribution by Sanne Bor to the writing of the paper possible.

**Conflicts of Interest:** Sanne Bor declares no conflict of interest. Steve Cropper is Academic Advisor to Partners in Paediatrics and has been since 1998. He is a member of the Core Group of the Partnership and attends the Board of Members as Academic Advisor. He received an annual fee from Partners in Paediatrics from 2000-2011, but since 2011 has undertaken the role *gratis*. He has published about the Partnership and its work on Managed Clinical Networks before, naming the Partnership with its consent, and will be preparing a history of the Partnership for publication, with past officers of the Partnership, again with the Partnership's approval and support.

**List of Annual Reports Partners in Paediatrics. (www.partnersinpaediatrics.org):**

AR1    Partners in Paediatrics (undated) Report of Year 1 of the Partnership. 60p
AR2    Partners in Paediatrics (undated) Report of Year 2 of the Partnership. 50p
AR3    Partners in Paediatrics (undated) Report of Year 3 of the Partnership. 39p
AR4    Partners in Paediatrics (undated) Report of Year 4 of the Partnership. 10p
AR5    Partners in Paediatrics (undated) Year five of the Partnership: Working to implement the National Service Framework. 12p
AR6    Partners in Paediatrics (undated) Year six of the Partnership: Working to implement change. 16p
AR7    Partners in Paediatrics (undated) Year seven of the Partnership: Competing priorities: What should ours be? 20pp
AR8    Partners in Paediatrics (2007/2008) Year eight of the Partnership: Working to implement the National Service Framework. 12p
AR9    Partners in Paediatrics (2008/09) Partners in Paediatrics Annual Report 2008/09. 12p
AR10   Partners in Paediatrics (2010) Annual Report 2010: Taking care of quality. 12p
AR11   Partners in Paediatrics (2011) Annual Report 2011: Helping to maintain and improve the quality of services for children and young people. 12p
AR12   Partners in Paediatrics (2014) PiP Report 2012–2014. Partners in Paediatrics. Solihull, West Midlands, 16p

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
