# Peer review of "(Un)bounding the Meta-Organization: Co-Evolution and Compositional Dynamics of a Health Partnership"

_admsci, doi:10.3390/admsci8030042_

Round 1
Reviewer 1 Report
This is an excellent paper that provides important insights for the theorizing of meta-organizations. The comparisons with other similar theories are also valuable. I have very little to add, but I beleive that the aim of the paper could be stated more clearly. In the end of section 2 the authors write that they are interested in looking at the dynamics at the external boundary both in terms of the external boundary and membership composition. Here, I think it would be appopriate to add that they are looking at changes in membership composition over a rather long period of time and that the insights generated by their study is due to their historial approach, and that their study demonstrates the importance of studying membership composition over time.
In section 4.2 there is an interesting idea about the relation between environmental contexts and the definition of similarity among members that could perhaps merit a little more discussion.
Reviewer 2 Report
(Un)bounding the meta-organization…
The introduction of this article is logical and well developed: purpose, definition of dynamics, contrast with prior studies, longitudinal case study, focus on patterning ( character: history, function, dynamics), appraisal of theoretical concepts and claims. Next, several relevant theoretical concepts/frameworks are introduced to characterize the chosen inter-organizational type ( association) and to present the selected investigation of dynamics in terms of institutional environment and membership composition. All these concepts and their interrelationships are succinctly introduced, questioned to some degree referring to other authors and theories whose alternative concepts are proposed without much guidance to the reader ( eg. Last part of pag 2, “ in their account of dynamics…). However, the main concepts ( institutional environment/ membership composition) are fully developed and detailed, so the reader can anticipate what to observe and what data are needed for the investigation.
Section Method & Data clearly gives an overview of raw data to “variables” in this study. Attention is given to the weaknesses in the data as well ( e.g. unnamed sources of tensions and frustration, therefore ambiguity in attribution…) . It also would be helpful if a short explanation was provided on how the four Phases were identified and on the basis of what criteria.
Section PIP, Environment and partnership (co)evolution. The phases are documented, analyzed and most of the time provided with some sense making interpretation ( eg. Facing incentives to competition, internal markets …). Exploring strategy and intentions besides organizational characteristics, composition and events feels mostly sufficiently in balance when reading.
Discussion and conclusion: the author(s) build their final appraisal of the theoretical statements, the partnership is a decided order but also an emergent order ( if one utilises a time frame …) and they show the interplay of boths sets of interrelations.
note: aabreviations FRO PiP could be introduced clearer
